# LiteVSR: Lightweight Adaptation of Frozen Diffusion Transformers for Video Super-Resolution

Yu Cao [1]   Ziquan Liu [1]   Zhensong Zhang [2]   Jiankang Deng [3]   Shaogang Gong [1]   Jifei Song [2]

## Abstract

Adapting large-scale pre-trained video generators for Video Super-Resolution (VSR) in novel domains remains computationally prohibitive. Methods that reformulate generation as direct Low-Quality to High-Quality mappings deviate from the original generative formulation, demanding extensive fine-tuning. ControlNet-style adapters lose their efficiency under modern Diffusion Transformers since the absence of encoder-decoder hierarchy forces duplication of the entire backbone. We observe that flow matching offers a principled alternative for cross-domain VSR adaptation. By predicting a constant velocity field across all timesteps, the adaptation task reduces to learning a fixed injection pattern rather than time-varying transformations. Building on this insight, we propose LiteVSR, a minimalist framework that performs VSR using a completely frozen Diffusion Transformer with a lightweight State-Aware Adapter. The adapter employs a dual-stream architecture that extracts static structural cues from the LQ input and dynamic cues from intermediate denoising states, aligning them through time-dependent cross-attention to enable adaptive transition from structural alignment to texture refinement as denoising proceeds. LiteVSR achieves competitive restoration quality with only 11.25% trainable parameters and 12 GPU-hours of training on a single A100, while maintaining fast sampling (down to a single step) compatibility.

## 1. Introduction

Video super-resolution (VSR) has undergone a fundamental paradigm shift in recent years, transitioning from fidelity-

[1]Queen Mary University of London [2]Huawei Darwin Research Center [3]Imperial College London. Correspondence to: Yu Cao <yu.cao@qmul.ac.uk>.

*Proceedings of the 43rd International Conference on Machine Learning*, Seoul, South Korea. PMLR 306, 2026. Copyright 2026 by the author(s).

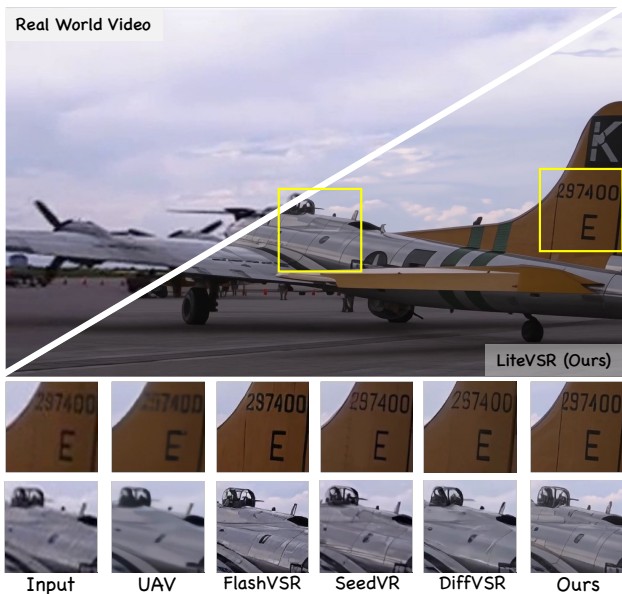

*Figure 1.* Visual comparisons of LiteVSR with SOTA methods (Zoom-in for best view).

*Table 1.* Training efficiency comparison. Percentages indicate trainable parameters within the diffusion backbone; additional fine-tuned VAE components are listed separately.

| Methods | Dataset | Trainable Params | Training Cost |
|---|---|---|---|
| UAV (Zhou et al., 2024) | WebVid-335K | ∼85% + Decoder | 32×A100, 80K iter |
| FlashVSR (Zhuang et al., 2025) | VSR-120K | 100% + Decoder | 32×A100, - |
| DiffVSR (Li et al., 2025) | WebVid-400K | 100% + Encoder | 8×A100, - |
| SeedVR (Wang et al., 2025b) | Private-5M | 100% + VAE | 32×H100, 115K iter |
| LiteVSR | REDS (266) | 11.25% | 1×A100, ∼6K iter |

oriented signal reconstruction to perception-driven detail synthesis (Blau & Michaeli, 2018; Rota et al., 2024). Traditional supervised VSR methods, trained on paired datasets with limited scale and diversity, struggle to generalize beyond their training distribution (Yang et al., 2021). In contrast, large-scale pre-trained video generators have learned rich priors about general natural video statistics from massive real-world data (Zhou et al., 2024; Chen et al., 2025). Recent efforts to leverage generative models for SR exploit a premise that such learned priors offer a promising foundation for realistic detail synthesis (Chan et al., 2022a).

Current Generative VSR methods fall into two categories: LQ-initialized and condition injection. The first directly learns LQ-to-HQ transformations (Zhuang et al., 2025;

Wang et al., 2025b; Chen et al., 2025), deviating from the original noise-to-video formulation and thus requiring extensive fine-tuning (Ho et al., 2020; Lipman et al., 2022). As shown in Table 1, this paradigm demands increasingly prohibitive resources as models scale, with recent methods requiring tens of A100/H100 GPUs and millions of training samples (Wang et al., 2025b; Zhuang et al., 2025; Li et al., 2025). Moreover, fine-tuning presents a fundamental **contradiction** as it inevitably degrades the pre-trained priors we aim to leverage (Ruiz et al., 2023; Zhong et al., 2024). **Condition injection** methods preserve the original generative process by treating low-quality inputs as conditioning signals. However, lightweight approaches such as LoRA (Hu et al., 2022) and feature concatenation (Yang et al., 2025; Tan et al., 2025) have poor control, failing to preserve structural fidelity to the input. ControlNet-style adapters (Zhang et al., 2023; Xie et al., 2025; Zhao et al., 2025) offer stronger control but lose their efficiency advantage under modern Diffusion Transformers. Without the encoder-decoder hierarchy, these methods must duplicate the entire backbone, resulting in parameter counts comparable to full fine-tuning and doubled memory consumption during inference (Peebles & Xie, 2023; Cao et al., 2025a). To solve the problem, we introduce an adaptation method that is both lightweight and capable of maintaining structural consistency with the low-quality input.

Unlike traditional Diffusion Model (Ho et al., 2020; Song et al., 2020), which predicts time-dependent noise or score functions, *flow matching* (Lipman et al., 2022) learns a constant velocity field toward clean data across all timesteps. This temporal consistency fundamentally simplifies the conditioning task: rather than learning time-varying transformations, the conditioning mechanism only needs to provide a fixed guidance signal at each DiT block. This property motivates a parameter-efficient design that keeps the generative backbone entirely frozen. As illustrated in Figure 2, our architecture processes two parallel branches through the same frozen DiT blocks: the main branch takes the noisy state $z_t$ for generation, while the condition branch extracts conditioning features through a lightweight adapter. At each DiT block, the condition branch features are projected into the main branch via a zero-initialized linear layer, providing structural guidance without disrupting the pretrained generation dynamics. Given this design, the adapter's role reduces to bridging a narrow gap between structural cues in the degraded input and the fine-grained details required for high-quality reconstruction, enabling effective adaptation with minimal trainable parameters.

Building on this insight, we propose LiteVSR, a minimalist framework that performs VSR with a completely frozen Diffusion Transformer and a lightweight State-Aware Adapter. A straightforward approach (Zhao et al., 2025) would inject structural information from the low-quality input by a

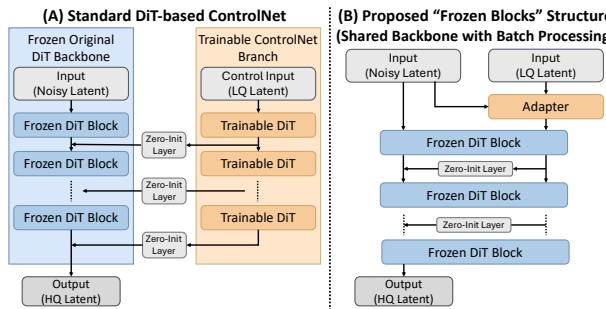

*Figure 2.* **ControlNet paradigms for DiT.** (A) Standard ControlNet duplicates the backbone for condition processing. (B) Our approach shares frozen DiT blocks via batch processing, requiring only a lightweight adapter.

fixed mapping, relying on a frozen generator to synthesize realistic details. However, this overlooks a key challenge: the optimal guidance signal should depend not only on the denoising timestep, but also on the current intermediate state (Zhang et al., 2023). As generation progresses, certain aspects of the reconstruction may already be well-formed while others remain deficient (Yue et al., 2024; Cao et al., 2025b). Effective conditioning requires sensing what the current estimate is missing and providing targeted guidance accordingly. This motivates our *state-aware* adapter design, which takes both the low-quality input and the evolving intermediate state as input, enabling it to adaptively modulate its guidance throughout the denoising process. To this end, our State-Aware Adapter employs a dual-stream architecture that jointly processes static cues from a low-quality input and dynamic cues from an evolving intermediate state. These two streams are fused via time-modulated cross-attention, where a learnable query attends to the concatenated features to extract the most relevant guidance at each denoising step.

We summarize our contributions as follows:

- Leverage the constant velocity prediction of flow matching to simplify VSR adaptation, enabling a completely frozen DiT backbone with only a lightweight adapter. To our knowledge, LiteVSR is the first framework that keeps all DiT blocks entirely frozen for VSR.

- Introduce a State-Aware Adapter with dual-stream processing and time-dependent cross-attention for adaptive structural-to-texture guidance during denoising.

- Achieve state-of-the-art quality with only 11.25% trainable parameters and 12 GPU-hours of training on a single A100. With off-the-shelf fast samplers, our method achieves competitive single-step generation on real-world benchmarks.

## 2. Related Work

### 2.1. Video Super Resolution

Traditional supervised VSR methods, including recurrent propagation frameworks (Isobe et al., 2020; Chan et al., 2021) and alignment-and-fusion architectures (Wang et al., 2019; Tian et al., 2020), learn restoration mappings from paired data. Early approaches rely on synthetic degradations such as bicubic downsampling (Nah et al., 2019), while recent work (Chan et al., 2022b; Yue et al., 2023; He et al., 2024) has shifted toward more realistic pipelines introduced by RealESRGAN (Wang et al.), which combines blur, noise, and compression to better approximate real-world conditions. Despite these advances in degradation modeling, supervised methods remain fundamentally constrained by the limited scale and diversity of high-resolution training data (Chen et al., 2025; Xie et al., 2025). This limitation has motivated the adoption of pre-trained video generators as powerful priors.

Existing approaches to leveraging generative priors fall into three categories: **Temporal modules on image diffusion models.** Upscale-A-Video (Zhou et al., 2024) integrates temporal layers with flow-guided latent propagation, MgLD-VSR (Yang et al., 2024a) introduces motion-guided attention, and UltraVSR (Liu et al., 2025) proposes degradation-aware scheduling. While these methods benefit from mature image priors, they inherit the limited temporal modeling of their base models. **ControlNet on video generators.** VEnhancer (He et al., 2024), STAR (Xie et al., 2025) and RealisVSR (Zhao et al., 2025) build video ControlNets (Zhang et al., 2023) on UNet-based backbones, achieving strong spatial and temporal quality. However, the transition from UNet to Diffusion Transformer (Peebles & Xie, 2023) in modern video generators disrupts this paradigm, as the absence of encoder-decoder hierarchy forces adapters like RealisVSR (Zhao et al., 2025) to replicate large portions of the backbone. **Fine-tuning video generators.** Multi-step approaches explore various training strategies: DiffVSR (Li et al., 2025) adopts progressive learning to handle complex degradations, while SeedVR (Wang et al., 2025b) employs mixed image-video training with shifted window attention for arbitrary-resolution restoration. To improve efficiency, recent work pursues one-step generation: DOVE (Chen et al., 2025) uses two-stage latent-pixel training, FlashVSR (Zhuang et al., 2025) applies three-stage distillation for streaming inference, and SeedVR (Wang et al., 2025b;a) leverages adversarial post-training.

The closest work to ours is OMGSR (Wu et al., 2025), which observed that mid-timestep latent distributions align well with low-quality inputs and accordingly injects LQ latents at a pre-computed timestep. However, this represents a static, one-time alignment that does not adapt as denoising progresses. Denoising is inherently dynamic (Preechakul

et al., 2022; Yue et al., 2024; Cao et al., 2025b): early steps benefit from structural information while later steps require fine-grained textures. Our proposed method learns an adaptive alignment that continuously adjusts throughout denoising, all while keeping the generator entirely frozen.

### 2.2. Video Diffusion Model

Early video diffusion models maintain explicit separation between spatial and temporal modeling. Some leverage pre-trained image diffusion backbones by inserting temporal modules, such as AnimateDiff (Guo et al., 2023) which adds motion modules to Stable Diffusion. Others train from scratch with dedicated spatial and temporal attention layers, as in Open-Sora's Spatial-Temporal Diffusion Transformer (STDiT) (Zheng et al., 2024). The adoption of Diffusion Transformers (DiT) (Peebles & Xie, 2023) and 3D positional encodings such as 3D RoPE (Su et al., 2024; Wei et al., 2025) has enabled unified architectures that jointly process spatial and temporal information without explicit separation. Representative models include CogVideoX (Yang et al., 2024b), which employs 3D VAE with full spatiotemporal attention, and HunyuanVideo (Kong et al., 2024), a 13B-parameter model with 3D causal VAE. Both adopt diffusion objectives with v-prediction. In parallel, flow matching (Lipman et al., 2022) has emerged as an alternative formulation that learns straight trajectories between noise and data distributions. Wan2.1/2.2 (Wan et al., 2025) combines the DiT architecture with flow matching, achieving strong performance with models ranging from 1.3B to 14B parameters.

## 3. Method

### 3.1. Preliminaries

**Latent Diffusion Models.** Diffusion models (Ho et al., 2020; Song et al., 2020) learn to generate data by reversing a gradual noising process. Given data $x_0$, the forward process adds Gaussian noise:

$$q(x_t|x_0) = \mathcal{N}(x_t; \sqrt{\bar{\alpha}_t}x_0, (1 - \bar{\alpha}_t)I) \quad (1)$$

where $\bar{\alpha}_t$ is a monotonically decreasing noise schedule. A neural network $\epsilon_\theta$ is trained to predict the added noise:

$$\mathcal{L}_{DM} = \mathbb{E}_{t,x_0,\epsilon}\left[\|\epsilon_\theta(x_t, t) - \epsilon\|^2\right] \quad (2)$$

Modern image and video generators perform this process in a compressed latent space for efficiency (Rombach et al., 2022). Given an input video $x \in \mathbb{R}^{T \times H \times W \times C}$ with $T$ frames of spatial resolution $H \times W$, a pre-trained VAE encoder $\mathcal{E}$ maps it to a latent representation $z = \mathcal{E}(x) \in \mathbb{R}^{t \times h \times w \times c}$, where $t = T/r_t$, $h = H/r_s$, $w = W/r_s$, with $r_t$ and $r_s$ denoting temporal and spatial compression ratios respectively. A decoder $\mathcal{D}$ reconstructs the output via $\hat{x} = \mathcal{D}(z)$. The diffusion process then operates entirely on $z$.

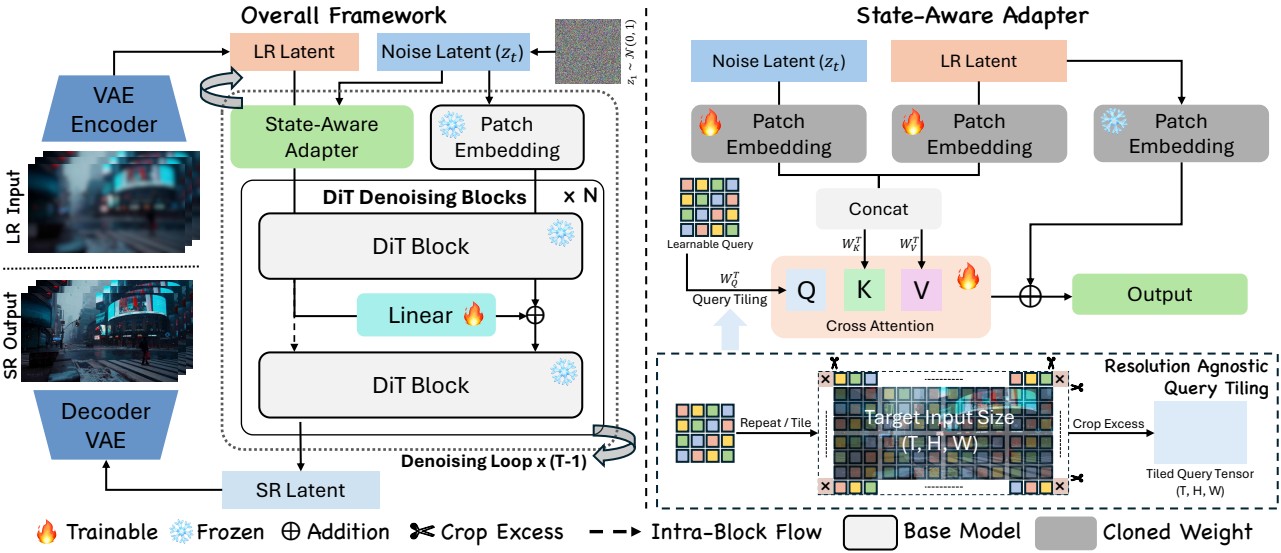

*Figure 3.* **LiteVSR.** *Left:* The overall framework keeps all DiT blocks frozen and injects control signals via zero-initialized linear layers. The State-Aware Adapter processes both the LR latent and the current noisy state to produce conditioning features. *Right:* The adapter employs dual-stream patch embeddings to extract features from the LR input and the denoising state, which are concatenated as keys and values. A learnable query attends to these features via cross-attention to produce the output. *Bottom:* Resolution-agnostic query tiling enables inference at arbitrary resolutions by repeating and cropping the learned query prototypes to match the target spatial dimensions.

**Flow Matching.** Our framework builds upon video generators trained with Flow Matching ([Lipman et al., 2022](#)), which formulates generation as learning a velocity field. Let $x_0 \sim q(x_0)$ be the data distribution and $x_1 \sim \mathcal{N}(0, I)$ be the prior. The probability path is defined as a linear interpolation $x_t = (1 - t)x_0 + tx_1$, where $t \in [0, 1]$. A neural network $v_\theta$ is trained to predict the velocity field:

$$\mathcal{L}_{FM} = \mathbb{E}_{t, x_0, x_1} \left[ \| v_\theta(x_t, t, c) - (x_1 - x_0) \|^2 \right] \quad (3)$$

where $c$ represents conditioning information. A key property of this formulation is that the target velocity $v = x_1 - x_0$ is constant across all timesteps, unlike the time-dependent noise scaling in DDPM. During inference, samples are generated by solving the ODE $dx_t/dt = v_\theta(x_t, t, c)$ from $t = 1$ to $t = 0$. At any timestep, the clean data can be estimated via $\hat{x}_{0,t} = x_t - (1 - t)v_\theta(x_t, t, c)$.

**Problem Definition.** Let $x \in \mathbb{R}^{T \times H \times W \times C}$ denote a high-quality video and $y = \Gamma(x)$ its degraded low-quality counterpart, where $\Gamma$ represents a degradation operator involving downsampling, blur, noise, and compression. The VSR problem seeks to recover $x$ from $y$. While degradation destroys fine details such as textures, it largely preserves structural information including layout and motion. Our goal is to leverage a pre-trained video generator to synthesize the missing details while maintaining structural consistency with the input.

### 3.2. LiteVSR Framework Overview

We propose LiteVSR, a lightweight VSR framework built upon frozen pre-trained video generators. The overall ar-

chitecture is illustrated in Figure 3. Given a low-quality video $y$, we encode it to latent space as $z_y = \mathcal{E}(y)$. At each denoising step, the generation process is formulated as:

$$z_{t-\Delta t} = z_t - \Delta t \cdot v_\theta(z_t, t, \mathcal{A}_\phi(z_y, \hat{z}_{0,t}, t)) \quad (4)$$

where $v_\theta$ is the frozen velocity network, $\hat{z}_{0,t}$ is the current clean estimate, and $\mathcal{A}_\phi$ is the proposed State-Aware Adapter that provides conditioning signals. This formulation offers a critical advantage over ControlNet-style adaptation. As shown in Figure 2, ControlNet requires a trainable backbone copy to process conditions, whereas our frozen backbone allows $z_y$ and $z_t$ to share the same DiT blocks via batched forward pass, eliminating parameter duplication and reducing memory consumption by nearly half. The remaining challenge is how to design $\mathcal{A}_\phi$ such that it provides sufficient control for VSR fidelity while remaining lightweight. We detail the adapter architecture in Sec. 3.3 and the training strategy in Sec. 3.4.

### 3.3. State-Aware Adapter

Unlike sparse conditions such as edges or poses, VSR demands strong fidelity to the input, making standard additive conditioning insufficient. Existing ControlNet-based VSR methods (*e.g.*, STAR, RealisVSR), thus discard the denoising state entirely, using only the low-quality input as conditioning. This leaves the adapter unaware of the evolving generation trajectory.

To address this, we design a State-Aware Adapter (Figure 3, right) $\mathcal{A}_\phi(z_y, \hat{z}_{0,t}, t)$ that takes three inputs: the low-quality latent $z_y = \mathcal{E}(y)$, the predicted clean estimate $\hat{z}_{0,t}$, and the

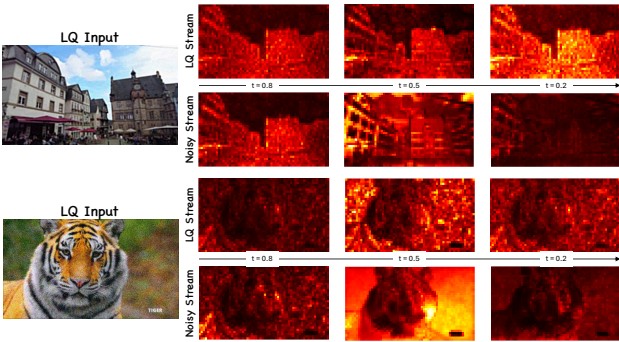

Figure 4. Attention maps illustrating the shift of focus across timesteps (t = 0.8, 0.5, 0.2) for the LQ stream and the noisy stream.

current timestep $t$. The core mechanism is a time-modulated cross-attention that dynamically balances structural fidelity and texture refinement:

$$C_{out} = \text{Attention}(Q_t, [K_{str} \oplus K_{ref}], [V_{str} \oplus V_{ref}]) \quad (5)$$

where $Q_t$ is a time-modulated query, $(K_{str}, V_{str})$ encode structural cues from the low-quality input, and $(K_{ref}, V_{ref})$ capture dynamic details from the current clean estimate.

**Dual-Stream Feature Projection.** We project both streams into a shared feature space $\mathbb{R}^{N \times D}$, where $N$ is the sequence length and $D$ is the feature dimension matching the DiT hidden size.

The *Structural Stream* extracts layout features $K_{str}$ from the low-quality input, serving as a static anchor:

$$K_{str} = \mathcal{F}_\phi^{str}(z_y) \quad (6)$$

The *Refinement Stream* extracts dynamic details $K_{ref} \in \mathcal{S}$ from the current clean estimate $\hat{z}_{0,t}$:

$$K_{ref} = \mathcal{F}_\phi^{ref}(\hat{z}_{0,t}) \quad (7)$$

where $\mathcal{F}_\phi^{str}$ and $\mathcal{F}_\phi^{ref}$ are learnable projection networks initialized from the base model's patch embedding to ensure feature compatibility. By using $\hat{z}_{0,t}$ instead of $z_t$, both streams operate within the clean data manifold, facilitating effective feature interaction. A residual connection is further added to prevent mode collapse and stabilize training.

**Resolution-Agnostic Time-Modulated Attention.** To handle inputs of arbitrary spatial-temporal resolution, we define the query as a small, learnable prototype window $Q_{win} \in \mathbb{R}^{1 \times h_w \times w_w \times D}$, where $h_w$ and $w_w$ denote the window size. This prototype is tiled across the input latent dimensions to match the sequence length $N$, enforcing translation invariance and enabling scalable inference. The tiled query is then modulated by the timestep $t$ via adaptive normalization (AdaLN) (Peebles & Xie, 2023):

$$Q_t = \text{Tile}(\gamma(t) \odot Q_{win} + \beta(t)) \quad (8)$$

This formulation enables the attention to function as a soft gate: at early stages ($t \to 1$), $Q_t$ attends primarily to structural features; as denoising progresses ($t \to 0$), attention

---

**Algorithm 1** LiteVSR Training and Inference

**Input:** frozen DiT $v_\theta$, VAE encoder $\mathcal{E}$, decoder $\mathcal{D}$, adapter parameters $\phi$
*// Training*
Sample $(x, y)$ from dataset, $t \sim p(t)$, $z_1 \sim \mathcal{N}(0, I)$
$z_0 \leftarrow \mathcal{E}(x)$, $z_y \leftarrow \mathcal{E}(y)$
$z_t \leftarrow (1-t)z_0 + tz_1$
$\hat{z}_0 \leftarrow z_y$          ▷ Initialize estimate with LQ latent
**for** $k = 1$ **to** $M(t)$ **do**
   $K_{str}, V_{str} \leftarrow \mathcal{F}_\phi^{str}(z_y)$     ▷ Static structural features
   $K_{ref}, V_{ref} \leftarrow \mathcal{F}_\phi^{ref}(\hat{z}_0)$     ▷ Dynamic refinement features
   $Q_t \leftarrow \text{Tile}(\gamma(t) \odot Q_{win} + \beta(t))$ ▷ Time-modulated query
   $c \leftarrow \text{Attention}(Q_t, [K_{str} \oplus K_{ref}], [V_{str} \oplus V_{ref}])$
   $\hat{z}_0 \leftarrow z_t - (1-t) \cdot v_\theta(z_t, t, c)$    ▷ Update clean estimate
**end for**
$\mathcal{L} \leftarrow \lambda(t) \|v_\theta(z_t, t, c) - (z_1 - z_0)\|^2$
*// Inference*
$z_1 \sim \mathcal{N}(0, I)$, $z_y \leftarrow \mathcal{E}(y)$, $\hat{z}_0 \leftarrow z_y$
**for** $t = 1 \to 0$ **with** step $\Delta t$ **do**
   Compute $c$ via adapter using $z_y$ and $\hat{z}_0$
   $z_{t-\Delta t} \leftarrow z_t - \Delta t \cdot v_\theta(z_t, t, c)$         ▷ Euler step
   $\hat{z}_0 \leftarrow z_{t-\Delta t} - (1 - t + \Delta t) \cdot v_\theta(z_{t-\Delta t}, t - \Delta t, c)$
**end for**
**Output:** $\mathcal{D}(z_0)$

---

shifts toward refinement features. In Figure 4, we visualize how the cross-attention dynamically adjusts its focus between the two streams as generation progresses.

### 3.4. Training Strategy

Unlike prior generative VSR methods that employ multi-stage training with pixel-space supervision (Chen et al., 2025; Zhuang et al., 2025), LiteVSR adopts a single-stage procedure optimized entirely in latent space. By operating solely with the flow matching objective, we eliminate the need for VAE decoding during training, significantly reducing memory footprint and accelerating convergence. Combined with our frozen backbone (83.72% of total parameters), this enables end-to-end training on a single A100 GPU using only 266 clips from REDS (Nah et al., 2019).

While our training pipeline is streamlined, it must still account for the iterative nature of the denoising process. We formulate the optimization to ensure robust learning across the entire diffusion trajectory through three components: recursive estimation, adaptive scheduling, and a weighted objective function.

**Recursive Refinement.** During inference, the model progressively refines its prediction using the output from the previous step. To align training with this behavior, we unroll the trajectory for $M$ steps to generate a refined condition:

$$\hat{z}_0^{(k)} = z_t - (1-t) \cdot v_\theta(z_t, t, \mathcal{A}_\phi(z_y, \hat{z}_0^{(k-1)}, t)) \quad (9)$$

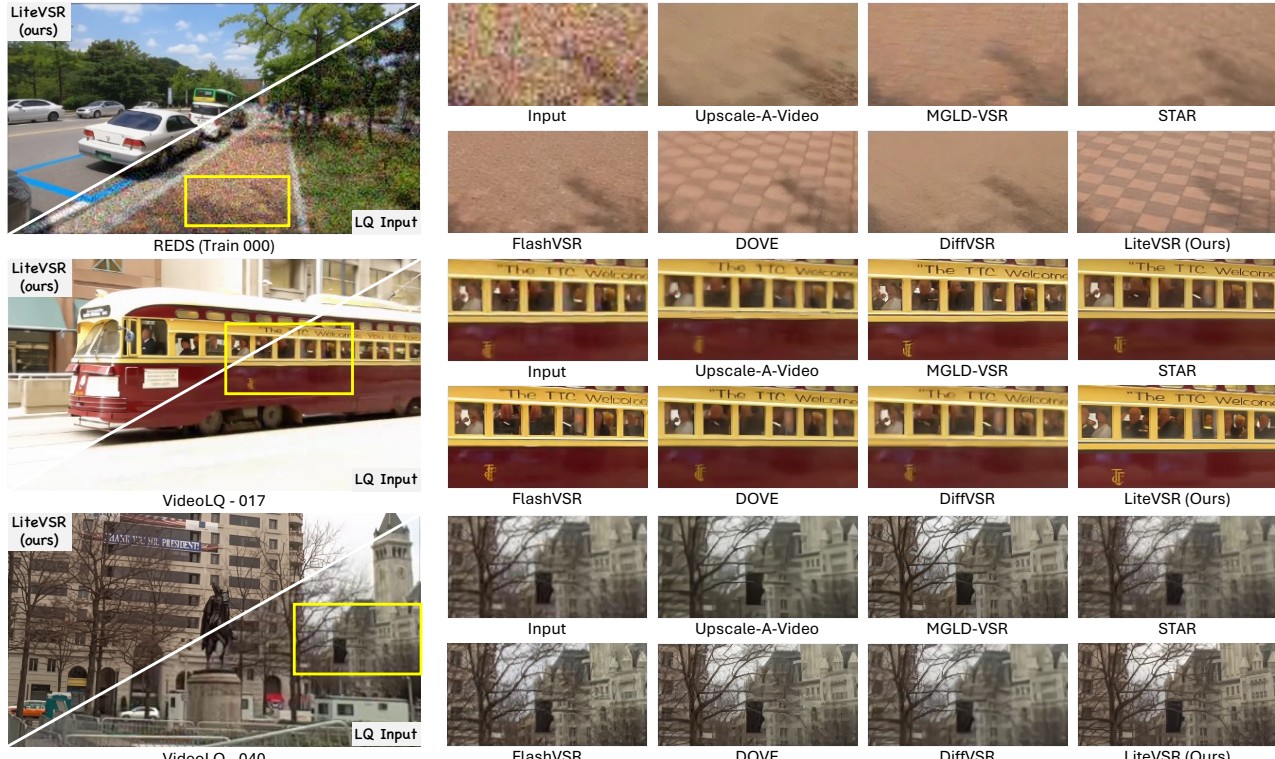

*Figure 5.* Qualitative comparison on REDS (first row) and VideoLQ (second and third row) datasets. **(Zoom in for best view)**

By initializing $\hat{z}_0^{(0)} = z_y$ and feeding the estimated $\hat{z}_0^{(k-1)}$ back into the adapter's refinement stream, we ensure that the attention mechanism learns to correct residual errors rather than suppressing the conditioning signal.

**Adaptive Trajectory Unrolling.** To balance computational efficiency with refinement quality, we employ a time-dependent schedule $M(t)$. Since fine-grained correction is most effective at low-noise states, we allocate more refinement steps as $t \to 0$. Specifically, we define the unroll depth using a shifted schedule:

$$M(t) = \left\lfloor 1 + \frac{s \cdot (1-t)}{1 + (s-1) \cdot (1-t)} \cdot (M_{max} - 1) \right\rfloor \quad (10)$$

where $s > 1$ controls the sharpness of the transition. This assigns minimal steps near $t = 1$ and increases nonlinearly as $t \to 0$. Following common practice in flow-based models, we set $s = 5$ (Esser et al., 2024; Wan et al., 2025).

**Training Objective.** We optimize the model using a weighted flow matching loss computed on the final unrolled estimate. Let $c_{ref} = \mathcal{A}_\phi(z_y, \hat{z}_0^{(M(t)-1)}, t)$ denote the refined conditioning signal derived from the adaptive trajectory. The total objective is defined as:

$$\mathcal{L} = \mathbb{E}_{t, z_0, z_1} \left[ \lambda(t) \left\| v_\theta(z_t, t, c_{ref}) - (z_1 - z_0) \right\|^2 \right] \quad (11)$$

where $\lambda(t)$ is a weighting function designed to prioritize

timesteps with high signal-to-noise ratios. We use $\lambda(t) = \sigma_t^{-2}$ in our experiments.

## 4. Experiment

**Datasets.** We train on the REDS dataset (Nah et al., 2019) with LR-HR pairs generated using the degradation pipeline of RealBasicVSR (Wang et al.). For evaluation, we consider both synthetic and real-world benchmarks. The synthetic sets include REDS4 (Nah et al., 2019), YouHQ40 (Zhou et al., 2024), UDM10 (Tao et al., 2017), and SPMCS (Yi et al., 2019), where LR frames are synthesized using the same degradation pipeline as training. We also evaluate on VideoLQ (Chan et al., 2022b), a real-world dataset containing diverse degradations without ground truth.

**Metrics and Baselines.** For datasets with ground truth, we report PSNR (Wang et al., 2004) as reference metrics, along with perceptual metrics including DISTS (Ding et al., 2020), LPIPS (Zhang et al., 2018), MUSIQ (Ke et al., 2021), NIQE (Mittal et al., 2012), CLIPIQA (Wang et al., 2023), and the video-specific metric DOVER (Wu et al., 2023), which measures both aesthetic quality and temporal consistency. For VideoLQ, we report only no-reference metrics (CLIP-IQA, DOVER, MUSIQ and NIQE). We compare against state-of-the-art approaches spanning different paradigms: Upscale-A-Video (Zhou et al., 2024), MGLD-VSR (Yang et al., 2024a), STAR (Xie et al., 2025) and DiffVSR (Li et al.,

*Table 2.* Quantitative comparison on REDS4, UDM10, SPMCS, YouHQ40 (synthetic), and VideoLQ (real-world). Best results are in **bold**; second-best are underlined.

| Datasets | Metrics | Upscale-A-Video | MGLD-VSR | STAR | FlashVSR | DOVE | DiffVSR | LiteVSR |
|---|---|---|---|---|---|---|---|---|
| REDS4 | PSNR ↑ | 20.2192 | 21.90 | 21.37 | 20.67 | **23.08** | 21.08 | 21.10 |
| | LPIPS ↓ | 0.4731 | 0.3190 | 0.4349 | 0.3202 | 0.3732 | 0.3677 | **0.3081** |
| | DISTS ↓ | 0.2539 | 0.1325 | 0.1763 | **0.1315** | 0.1982 | 0.1552 | 0.1359 |
| | CLIPIQA ↑ | 0.2042 | 0.2970 | 0.2045 | 0.3186 | 0.3017 | 0.2877 | **0.3748** |
| | DOVER ↑ | 0.2853 | 0.3376 | 0.3320 | 0.3451 | 0.3402 | 0.3019 | **0.3622** |
| | NIQE ↓ | 5.2102 | 3.5366 | 4.5904 | 2.9378 | 4.9108 | 3.1590 | **2.6938** |
| | MUSIQ ↑ | 39.9466 | 60.87 | 43.15 | 62.74 | 57.07 | 64.71 | **65.99** |
| UDM10 | PSNR ↑ | 22.76 | 23.96 | 24.15 | 23.32 | **25.74** | 22.34 | 23.01 |
| | LPIPS ↓ | 0.4246 | 0.3231 | 0.4069 | 0.2738 | **0.2759** | 0.3341 | 0.3266 |
| | DISTS ↓ | 0.2427 | 0.1533 | 0.2107 | **0.1354** | 0.1537 | 0.1799 | 0.164 |
| | CLIPIQA ↑ | 0.2515 | 0.4286 | 0.2214 | 0.4958 | 0.5348 | 0.355 | **0.558** |
| | DOVER ↑ | 0.2484 | 0.3899 | 0.227 | 0.4618 | 0.4673 | 0.44 | **0.515** |
| | NIQE ↓ | 6.3404 | 3.9219 | 6.0595 | 3.9426 | 5.1821 | 4.8054 | **3.8333** |
| | MUSIQ ↑ | 35.89 | 60.71 | 32.56 | 67.51 | 65.11 | 57.40 | **70.02** |
| SPMCS | PSNR ↑ | 19.09 | 20.78 | 20.44 | 20.33 | **21.75** | 19.93 | 19.76 |
| | LPIPS ↓ | 0.5230 | 0.4046 | 0.4826 | **0.3536** | 0.3682 | 0.4232 | 0.3808 |
| | DISTS ↓ | 0.3151 | 0.2074 | 0.2546 | 0.1949 | 0.1973 | 0.2978 | **0.1917** |
| | CLIPIQA ↑ | 0.3190 | 0.4616 | 0.3206 | 0.4823 | 0.5681 | 0.4021 | **0.5726** |
| | DOVER ↑ | 0.2126 | 0.3091 | 0.2745 | 0.4065 | 0.3800 | 0.3448 | **0.4093** |
| | NIQE ↓ | 5.7175 | 3.7654 | 5.7116 | 3.5318 | 4.9439 | 4.5756 | **3.4324** |
| | MUSIQ ↑ | 41.52 | 65.41 | 44.72 | 70.33 | 69.83 | 67.24 | **70.42** |
| YouHQ40 | PSNR ↑ | 20.99 | 22.12 | 22.66 | 21.21 | **23.67** | 20.59 | 21.28 |
| | LPIPS ↓ | 0.4964 | 0.3781 | 0.4747 | **0.3049** | 0.3377 | 0.3909 | 0.3842 |
| | DISTS ↓ | 0.2529 | 0.1570 | 0.2120 | **0.1248** | 0.1639 | 0.1854 | 0.1816 |
| | CLIPIQA ↑ | 0.2846 | 0.4413 | 0.2560 | 0.5278 | 0.4919 | 0.3976 | **0.5741** |
| | DOVER ↑ | 0.3747 | 0.5019 | 0.3521 | 0.5766 | 0.5805 | 0.4769 | **0.5984** |
| | NIQE ↓ | 6.5980 | 3.6783 | 6.3965 | 3.8682 | 4.9591 | 4.7449 | **3.5094** |
| | MUSIQ ↑ | 31.40 | 59.33 | 27.67 | **69.51** | 62.86 | 55.60 | 68.67 |
| VideoLQ | CLIPIQA ↑ | 0.2496 | 0.4524 | 0.26288 | 0.4236 | 0.3228 | 0.2895 | **0.4681** |
| | DOVER ↑ | 0.3107 | 0.3389 | 0.3961 | **0.5037** | 0.4592 | 0.4202 | 0.4846 |
| | NIQE ↓ | 6.0349 | 3.8245 | 6.2112 | 3.8623 | 5.3030 | 4.7311 | **3.76** |
| | MUSIQ ↑ | 27.07 | 49.07 | 33.94 | 56.14 | 44.69 | 44.9420 | **59.05** |

2025) (multi-step diffusion), and DOVE (Chen et al., 2025) and FlashVSR (Zhuang et al., 2025) (one-step diffusion).

**Implementation Details.** We implement LiteVSR in PyTorch using Wan2.2-5B (Wan et al., 2025) as the base video generator. Unlike prior methods that require text captions, we use an empty text prompt pre-encoded to reduce inference overhead. Training videos are randomly cropped to $512 \times 512$ resolution. We freeze all DiT blocks and train only the proposed State-Aware Adapter along with a lightweight linear fusion layer that combines the adapter output with the DiT features. The model is optimized using the flow matching objective (L2 loss) (Lipman et al., 2022) in latent space, without any pixel-domain loss. We use the AdamW optimizer (Loshchilov & Hutter, 2019) with constant learning rate $5 \times 10^{-5}$, $\beta_1 = 0.9$, $\beta_2 = 0.999$, and weight decay 0.01. We train for 6,250 iterations on a single A100 GPU with batch size 1 and gradient accumulation over 8 steps. Total training time is approximately 12 GPU-hours. Further implementation details are provided in Appendix B.

## 4.1. Results

**Quantitative Analysis** We compare LiteVSR against state-of-the-art VSR methods on both synthetic (REDS4, UDM10, SPMCS, YouHQ40) and real-world (VideoLQ) benchmarks. As shown in Table 2, our method achieves the best performance on perceptual metrics across most datasets. This indicates that LiteVSR generates results with superior perceptual quality and naturalness. Notably, LiteVSR achieves dominant performance on REDS4, the dataset used for training, while also obtaining the best results on VideoLQ, a real-world benchmark with unseen degradations. This demonstrates strong intra-domain restoration capability as well as robust cross-domain generalization. Since our backbone remains entirely frozen, adapting to new domains requires only retraining the lightweight adapter, enabling practical deployment across diverse real-world scenarios.

**Qualitative Analysis** Figure 5 presents visual comparisons on REDS (in-domain) and VideoLQ (cross-domain) examples. For clarity, we enlarge selected local patches to better illustrate the differences among all methods. Overall,

LiteVSR produces sharper and more faithful reconstructions, while competing methods tend to fill in missing details with artifacts rather than recovering the actual content. For example, in the brick pavement scene under heavy degradation (First row), LiteVSR successfully recovers straight, well-defined edges, whereas other methods either produce blurry results (Upscale-A-Video, STAR) or over-smooth the structure entirely (DOVE). This demonstrates the advantage of leveraging frozen generative priors: rather than memorizing texture templates, the model synthesizes contextually appropriate details. LiteVSR also exhibits superior temporal consistency, stably recovering text and patterns on a fast-moving bus (Second row) across frames where other methods produce flickering artifacts.

In regions with high information density, such as distant scenes or dense textures, super-resolution becomes increasingly challenging. The third row presents such a case: DOVE and FlashVSR restore some local details but introduce noticeably unnatural artifacts. Figure 6 further examines this with greenery and hair, where fully fine-tuned methods produce grainy, unrealistic textures, while LiteVSR generates more coherent details. Additional video comparisons are provided in the supplementary material.

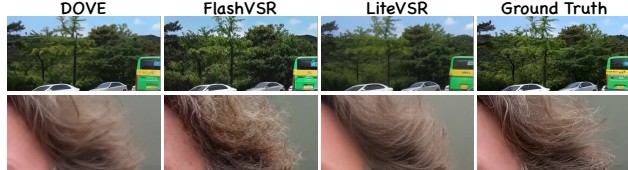

*Figure 6.* Visual comparison on high-density detail regions (greenery and hair).

**User Study.** We conduct a user study with 15 participants on 17 sequences against DOVE and FlashVSR. The sequences cover three scenarios: 5 clips from VideoLQ (Standard) and 12 real-world videos grouped into Simple and Extreme by their inherent quality. Each sequence is presented with randomized A/B/C assignment. As shown in Table 3, LiteVSR is consistently preferred across all metrics, attaining 75.5% overall preference and 70.0% on temporal consistency despite operating on a frozen backbone. The advantage grows monotonically with degradation severity, reaching 91.2% under extreme conditions, indicating stronger cross-domain robustness than fully fine-tuned baselines.

### 4.2. Ablation Study

We investigate the effectiveness of the proposed Adaptive Unrolling training strategy and corresponding Hyperparameter selection.

**Dual-Stream Design.** We ablate three variants of the State-Aware Adapter: *w/o Refinement Stream* conditions only on $z_y$; *w/ Noisy Latent* replaces $\hat{z}_{0,t}$ with $z_t$ in the refinement stream; *w/o Time Modulation* removes both dual-stream

*Table 3.* User study results. The Overall block aggregates across all 17 sequences with three evaluation metrics; the per-scenario block breaks down overall preference by input video quality. Values indicate the percentage of participants preferring each method.

| Scenario | Metric | DOVE | FlashVSR | LiteVSR |
|---|---|---|---|---|
| Overall | Visual Quality | 5.8% | 18.7% | **75.5%** |
| | Temporal Consistency | 9.3% | 20.6% | **70.0%** |
| | Overall Preference | 6.2% | 18.3% | **75.5%** |
| Standard | Overall Preference | 5.3% | 36.0% | **58.7%** |
| Simple | Overall Preference | 6.6% | 19.8% | **73.6%** |
| Extreme | Overall Preference | 6.6% | 2.2% | **91.2%** |

*Table 4.* Ablation on the dual-stream adapter design (evaluated on VideoLQ).

| Variant | CLIPIQA↑ | NIQE↓ | DOVER↑ | MUSIQ↑ |
|---|---|---|---|---|
| w/o Refinement Stream | 0.4603 | 3.7782 | **0.4878** | 58.64 |
| w/ Noisy Latent | 0.4570 | 3.7804 | 0.4875 | 58.44 |
| w/o Time Modulation | 0.4292 | 4.3252 | 0.4663 | 52.22 |
| Full (✓) | **0.4681** | **3.7600** | 0.4846 | **59.05** |

inputs and the time-modulated query. As shown in Table 4, removing time modulation causes the largest degradation, as static conditioning cannot adapt to the evolving denoising trajectory. Replacing $\hat{z}_{0,t}$ with $z_t$ disrupts feature interaction since the noisy and clean latents lie in different distributions, while omitting the refinement stream yields timestep-invariant guidance and leaves fine details under-restored.

**Effectiveness of the Adaptive Unrolling Strategy**. Our State-Aware Adapter takes both the low-quality latent $z_y$ and a clean estimate $\hat{z}_0$ as input. In standard flow matching training, $\hat{z}_0$ is not accessible since $z_t$ is directly constructed via interpolation without model inference. However, at test time the adapter must process predicted estimates from the model itself, creating a train-test mismatch. The Adaptive Unrolling Strategy (AUS) bridges this gap by unrolling the model during training to produce $\hat{z}_0$, exposing the adapter to realistic intermediate states. Table 5 (first block) validates this design. Without AUS, the adapter overfits to ground truth conditioning and struggles at inference time. Enabling AUS yields consistent improvements with only ~14% additional training cost.

**Window Size for Learnable Query** As introduced in Sec. 3.3, we employ a learnable query prototype $Q_{win} \in \mathbb{R}^{1 \times h_w \times w_w \times D}$ that is tiled to match arbitrary input resolutions. The window size $(h_w, w_w)$ governs a trade-off between receptive field and generalization. A larger window increases context but reduces exposure to tiling during training; a smaller window ensures tiling generalization but limits receptive field. We evaluate three window sizes: $32 \times 32$ (covering the full $512 \times 512$ pixel crop), $16 \times 16$, and $8 \times 8$, corresponding to progressively smaller receptive fields. As shown in the second block of Table 5, the $32 \times 32$ configuration underperforms despite more learnable param-

*Table 5.* Ablation studies on VideoLQ. We evaluate sampling steps, query window size, injection layer rank, and the adaptive unrolling strategy (AUS). Checkmarks (✓) indicate the default settings used in Table 2.

| Ablation | Setting | CLIPIQA↑ | NIQE↓ | DOVER↑ | MUSIQ↑ |
|---|---|---|---|---|---|
| Adaptive | w/o AUS | 0.4430 | 4.0487 | 0.4805 | 56.30 |
| Unrolling (AUS) | w/ AUS (✓) | **0.4642** | **3.7898** | **0.4849** | **58.62** |
| | 8 × 8 | 0.4549 | 3.7908 | 0.4823 | 58.20 |
| Window Size | 16 × 16 (✓) | **0.4642** | **3.7898** | 0.4849 | **58.62** |
| | 32 × 32 | 0.4587 | 3.7943 | **0.4850** | 58.57 |
| | 1 steps | 0.4522 | 4.2565 | 0.4454 | 57.01 |
| Sampling Steps | 5 steps (✓) | **0.4642** | 3.7898 | 0.4849 | **58.62** |
| | 10 steps | 0.4589 | **3.6741** | 0.4911 | 58.44 |
| | 15 steps | 0.4383 | 3.6908 | **0.4934** | 57.57 |
| | Full Rank (✓) | 0.4642 | 3.7898 | **0.4849** | **58.62** |
| Injection Rank | LoRA-128 | **0.4693** | **3.7304** | 0.4748 | 58.50 |
| | LoRA-64 | 0.4621 | 3.7887 | 0.4700 | 57.70 |

eters, as it never encounters tiling during training. The $8 \times 8$ window suffers from limited receptive field. A $16 \times 16$ window ($256 \times 256$ pixels) strikes the optimal balance.

**Computational Efficiency and Fast Sampling.** Our design introduces minimal computational overhead: the adapter adds only ∼50ms per step, while the parallel condition branch increases inference time by approximately 8% on an A100 GPU at $512 \times 512$ resolution. By preserving the original flow matching formulation, LiteVSR naturally supports arbitrary sampling steps without additional distillation. As shown in the third block of Table 5, we evaluate with 5, 10, and 15 steps using the UniPC scheduler (Zhao et al., 2023). Performance scales consistently with step count, while even 5-step sampling yields competitive quality. Notably, single-step generation without any distillation already achieves comparable results to DOVE and FlashVSR. This confirms that our adapter injection does not disrupt the underlying ODE trajectory, enabling flexible quality-speed trade-offs at inference time.

**Further Parameter Compression.** We investigate whether the injection layers can be further compressed via low-rank adaptation (LoRA) (Hu et al., 2022). As shown in the fourth block of Table 5, replacing full-rank projection with LoRA-128 reduces trainable parameters by 40.9% (634M → 375M) while achieving comparable or even superior performance. LoRA-64 further reduces parameters by 42.7% (634M → 363M) with only marginal degradation. This suggests that the conditioning signal has low intrinsic dimensionality, which aligns with our hypothesis that flow matching's constant velocity field simplifies the injection pattern.

## 5. Conclusion

We presented LiteVSR, a lightweight framework that achieves competitive video super-resolution quality while requiring only 11.25% trainable parameters and minimal training data. In practice, no single VSR model generalizes across all domains, necessitating frequent retraining for different content types or degradation patterns. By keeping

the generative backbone entirely frozen, LiteVSR enables rapid domain adaptation on consumer hardware, making it practical to customize high-quality restoration models for diverse real-world deployment scenarios.

## Impact Statement

This paper presents work whose goal is to advance the field of Machine Learning. There are many potential societal consequences of our work, none which we feel must be specifically highlighted here.

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

## A. Appendix Overview

This is the appendix for "LiteVSR: Lightweight Adaptation of Frozen Diffusion Transformers for Video Super-Resolution". Tab. 7 summarizes the abbreviations and symbols used in the paper.

This appendix is organized as follows:

- Section B presents additional implementation details of our approach.

- Section C provides additional qualitative comparisons in video format.

- Section D discusses the limitation of our work.

## B. Implementation Detail

**Inference Details.** For all benchmarks, we use 5 sampling steps with the UniPC scheduler (Zhao et al., 2023) from Wan2.2 (Wan et al., 2025) with default setting. REDS4 consists of clips 000, 011, 015, and 020 from the REDS (Nah et al., 2019) training set. For VideoLQ, we apply spatial tiling due to the memory footprint of the VAE decoder. Image quality metrics (CLIPIQA, NIQE, MUSIQ, LPIPS, DISTS) are computed using PyIQA (Chen & Mo, 2022) with default settings. For DOVER, we follow the official implementation from the original paper (Wu et al., 2023). Other Implementation detail are listed in Table 6.

*Table 6.* Implementation details and hyperparameters

| Configuration | Value | Configuration | Value |
|---|---|---|---|
| **Model Architecture** | | **Training Settings** | |
| Base Model | Wan2.2-5B (Wan et al., 2025) | Training Dataset | REDS (Nah et al., 2019) |
| Total Parameters | 5.6B | Training Resolution | 37 x 512 x 512 |
| Trainable Parameters | 634M | Batch Size | 1 |
| Query Window Size $(h_w, w_w)$ | (1, 16, 16) | Gradient Accumulation Steps | 8 |
| | | Total Iterations | 6250 |
| | | Training Time | $\sim$12 GPU (A100) Hour |
| **Optimizer** | | **Training Strategy** | |
| Optimizer | AdamW (Loshchilov & Hutter, 2019) | Max Unrolling Depth $M_{max}$ | 3 |
| Learning Rate | $5 \times 10^{-5}$ | Schedule Sharpness $s$ | 5 |
| Learning Rate Schedule | Constant | Loss Weighting $\lambda(t)$ | $\sigma_t^{-2}$ |
| $\beta_1, \beta_2$ | 0.9, 0.999 | | |
| Weight Decay | 0.01 | | |

## C. Additional Qualitative Results

We provide video comparisons in the supplementary material to better demonstrate temporal consistency and visual quality. Each video presents side-by-side comparisons of FlashVSR, DOVE, and our LiteVSR on the VideoLQ benchmark. Due to file size constraints, the supplementary videos are compressed and limited to shorter sequences; uncompressed results for all test samples will be released upon publication.

## D. Limitation

While LiteVSR achieves strong performance on natural scenes, buildings, and human subjects, it shares a common limitation with other generative restoration methods: the inability to faithfully reconstruct text content. As shown in Figure 7, when super-resolving videos containing text such as book covers, street signs, or billboards, the model tends to generate plausible but incorrect characters, especially under severe degradation where structural cues become ambiguous. This is an inherent challenge for generative approaches, as they lack explicit linguistic priors to constrain text synthesis. Future work may explore integrating OCR-guided constraints or text-aware modules to address this limitation.

*Table 7.* List of abbreviations and symbols used in the paper

| Symbol / Abbr. | Meaning |
| --- | --- |
| **Video and Latent Space Symbols** | |
| $x$ | High-quality video, $x \in \mathbb{R}^{T \times H \times W \times C}$ |
| $y$ | Low-quality (degraded) video |
| $\Gamma$ | Degradation operator (downsampling, blur, noise, compression) |
| $z, z_0$ | Latent representation of clean data |
| $z_1$ | Sampled noise from $\mathcal{N}(0, I)$ |
| $z_t$ | Interpolated latent at timestep $t$: $(1-t)z_0 + tz_1$ |
| $z_y$ | Latent representation of LQ video: $\mathcal{E}(y)$ |
| $\hat{z}_{0,t}, \hat{z}_0$ | Predicted clean estimate from noisy state |
| $\mathcal{E}, \mathcal{D}$ | VAE encoder, VAE decoder |
| $T, H, W, C$ | Number of frames, height, width, channels |
| $t, h, w, c$ | Compressed latent dimensions |
| $r_t, r_s$ | Temporal and spatial compression ratios |
| **Diffusion and Flow Matching Symbols** | |
| $q(x_t\|x_0)$ | Forward process distribution |
| $\bar{\alpha}_t$ | Cumulative noise schedule parameter |
| $\epsilon_\theta$ | Noise prediction network |
| $v_\theta$ | Velocity field network (flow matching) |
| $t$ | Timestep, $t \in [0, 1]$ |
| $\Delta t$ | Timestep interval for sampling |
| $c$ | Conditioning information |
| $\mathcal{L}_{DM}$ | Diffusion model loss |
| $\mathcal{L}_{FM}$ | Flow matching loss |
| **State-Aware Adapter Symbols** | |
| $\mathcal{A}_\phi$ | State-Aware Adapter with parameters $\phi$ |
| $\phi$ | Learnable adapter parameters |
| $\theta$ | Frozen DiT backbone parameters |
| $K_{str}, V_{str}$ | Keys and values from structural stream (LQ input) |
| $K_{ref}, V_{ref}$ | Keys and values from refinement stream (clean estimate) |
| $\mathcal{F}_\phi^{str}$ | Structural stream projection network |
| $\mathcal{F}_\phi^{ref}$ | Refinement stream projection network |
| $Q_t$ | Time-modulated query |
| $Q_{win}$ | Learnable query prototype window, $Q_{win} \in \mathbb{R}^{1 \times h_w \times w_w \times D}$ |
| $h_w, w_w$ | Query window height and width |
| $N$ | Sequence length |
| $D$ | Feature dimension (matching DiT hidden size) |
| $C_{out}$ | Cross-attention output |
| $\oplus$ | Concatenation operator |
| **Training Strategy Symbols** | |
| $M, M(t)$ | Unrolling depth / number of refinement steps |
| $M_{max}$ | Maximum unrolling depth |
| $s$ | Schedule sharpness parameter (default: 5) |
| $\hat{z}_0^{(k)}$ | Clean estimate at $k$-th unrolling iteration |
| $c_{ref}$ | Refined conditioning signal after adaptive unrolling |
| $\lambda(t)$ | Loss weighting function, $\lambda(t) = \sigma_t^{-2}$ |
| $\sigma_t$ | Noise level at timestep $t$ |
| $\mathcal{L}$ | Total training objective |

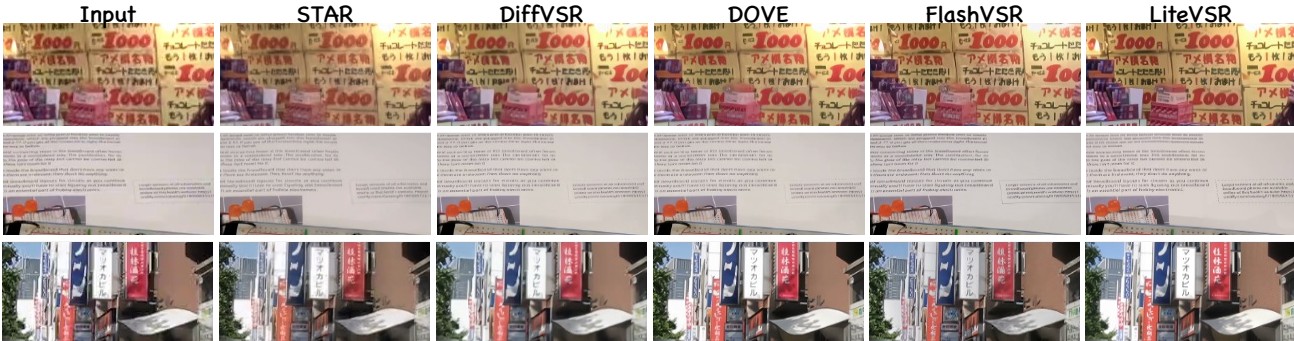

*Figure 7.* Limitation of generative VSR methods on text reconstruction. All methods, including ours, struggle to faithfully restore text content under degradation, often generating plausible but incorrect characters.

