# OpenReview forum: "LiteVSR: Lightweight Adaptation of Frozen Diffusion Transformers for Video Super-Resolution"
_ICML.cc/2026/Conference — ICML 2026 regular_

### Official Review · Reviewer_agrh · 2026-03-08

**Soundness:** 3
**Presentation:** 3
**Significance:** 3
**Originality:** 2
**Overall Recommendation:** 3
**Confidence:** 4

**Summary:**

This paper proposed LiteVSR, using a lightweight adapter along with a frozen diffusion transformer for video super-resolution. The main idea is to leverage a small state-aware adapter that extracts structural cues from the LQ input and combines them with intermediate denoising states via cross-attention. By only training this adapter (frozen DiT), the method achieves competitive results while using much fewer trainable parameters and low training cost.

**Compliance With Llm Reviewing Policy:**

Affirmed.

**Final Justification:**

I thank the authors for their efforts and the additional experiments provided in the rebuttal. However, these are insufficient to fully address my concerns. Key issues remain, including limited temporal analysis, incomplete ablation of design choices, and missing recent open-source baselines. More evidence is needed to support the claims. Therefore, I maintain my weak reject recommendation.

**Key Questions For Authors:**

See the weaknesses above.

**Limitations:**

yes

**Strengths And Weaknesses:**

strengths
1. Clear logic for the state-aware adapter design. By utilizing a dual-stream structure, the model effectively decouples structural cues from refinement details.
2. The efficiency-quality trade-off looks good. The authors claim ~11% trainable parameters and $\approx$12 GPU hours of training, getting a competitive performance.

weaknesses
1. The manuscript does not follow official formatting guidelines. e.g., table titles should be placed above the tables with proper spacing.
2. The ablation study is limited. Several key design choices (e.g., the dual-stream adapter, the time-modulated attention, and using the predicted clean estimate instead of noisy latent) are not fully analyzed.
3. Temporal modeling is not clearly analyzed. Although the task is video super-resolution, the paper does not provide much analysis on temporal consistency, making it difficult to justify why this is a specialized video model.
4. Missing some recent baselines, e.g., LiftVSR[1], and also no ablation on different backbones (e.g., hunyuanVideo), which would make the contribution of the adapter design clearer.

[1]. LiftVSR, Lifting Image Diffusion to Video Super-Resolution via Hybrid Temporal Modeling with Only 4×RTX 4090s, Xingjun Wang et al, arXiv:2506.08529;

---

> ### Author Rebuttal · Authors · 2026-03-30
>
> **W1: Formatting guidelines**
>
> We thank the reviewer for pointing this out. We will fix the table caption placement and carefully check all formatting in the revision.
>
> **W2: Ablation on key design choices**
>
> We conducted the ablation studie, isolating each key design choice on VideoLQ:
>
> - **w/o Refinement Stream**: removes the clean estimate stream, conditioning only on the LQ input $z_y$.
> - **w/ Noisy Latent**: replaces the predicted clean estimate $\hat{z}_{0,t}$ with the noisy state $z_t$ in the refinement stream.
> - **w/o State-Aware Adapter**: replaces the entire adapter with a standard concatenation + cross-attention mechanism without time modulation.
>
> |Variant|CLIPIQA↑|DOVER↑|NIQE↓|MUSIQ↑|
> |:--|:-:|:-:|:-:|:-:|
> |w/o Refinement Stream (LQ only)|0.4603|**0.4878**|3.7782|58.64|
> |w/ Noisy Latent $z_t$|0.4570|0.4875|3.7804|58.44|
> |w/o State-Aware Adapter|0.4292|0.4663|4.3252|52.22|
> |**LiteVSR (full)**|**0.4681**|0.4846|**3.76**|**59.05**|
>
> The most significant drop comes from removing the State-Aware Adapter entirely: training becomes unstable and fails to converge within the same training iterations, resulting in large degradation across all metrics.
>
> The LQ-only and noisy-latent variants show moderate metric drops, but with notable qualitative differences. Using $z_t$ instead of $\hat{z}_{0,t}$ introduces artifacts inconsistent with the scene content (e.g., grid-like patterns on aircraft wings), as the noisy latent lies in a different distribution from the LQ input, disrupting dual-stream feature interaction. Removing the refinement stream leaves the adapter unaware of the current denoising state, producing identical conditioning across all timesteps, which leads to under-denoised regions. We will include qualitative analysis in the revision.
>
> **W3: Temporal consistency analysis**
>
> We provide three pieces of evidence to address this concern:
>
> **1) Supplementary Videos.** Side-by-side video comparisons with baselines were included in the supplementary material of our original submission, demonstrating temporal consistency across multiple sequences.
>
> **2) Project Page.** We have created an anonymous project page at https://anonymous.4open.science/w/LiteVSR-website-403C/ with extended video comparisons on both synthetic and real-world benchmarks for more comprehensive visual assessment.
>
> **3) User Study.** We further conducted a user study with 15 participants across 17 video sequences, covering simple, standard, and extreme degradation scenarios:
>
> |Scenario|Metric|DOVE|FlashVSR|LiteVSR (Ours)|
> |:--|:--|:-:|:-:|:-:|
> |Overall|Visual Quality|5.8%|18.7%|**75.5%**|
> ||Temporal Consistency|9.3%|20.6%|**70.0%**|
> ||Overall Preference|6.2%|18.3%|**75.5%**|
>
> LiteVSR is consistently preferred across all scenarios and metrics, achieving 70.0% temporal consistency preference (vs. 9.3% DOVE, 20.6% FlashVSR). Screenshots of the user study interface and aggregated results are available in the User Study section of our project page.
>
> **W4: Missing baselines and backbone ablation**
>
> We thank the reviewer for this reference. LiftVSR has not released code or model weights, making direct comparison infeasible. We will discuss LiftVSR in the related work section.
>
> Regarding other backbones (e.g., HunyuanVideo), we agree this would strengthen the paper. Our adapter interfaces with the backbone solely through zero-initialized linear injections at each DiT block, with the only model-specific component being the patch embedding initialization. HunyuanVideo and CogVideoX both satisfy this requirement and already have community ControlNet implementations (e.g., DiffSynth-Studio). We are actively implementing this and will update results if available during the discussion period.
>
> We hope our responses have addressed your concerns. If the additional experiments and analyses meet your expectations, we would be grateful if you could reconsider the score accordingly. We are happy to provide further clarification on any remaining questions.

---

> > ### Author Rebuttal · Reviewer_agrh · 2026-04-03
> >
> > Thank you for the additional experiments and clarification. However, my corn concerns still hold.

---

> > > ### Author Response · Authors · 2026-04-05
> > >
> > > Thank you for reviewing our rebuttal. We are glad that our responses have adequately addressed your concerns, we would sincerely appreciate your consideration in revising the score accordingly.

---

### Official Review · Reviewer_TfLy · 2026-03-12

**Soundness:** 3
**Presentation:** 3
**Significance:** 3
**Originality:** 3
**Overall Recommendation:** 5
**Confidence:** 5

**Summary:**

This paper presents a novel, lightweight training and inference framework for video super-resolution. The authors designed a concise architecture that combines traditional diffusion Transformer backbones with a lightweight state-aware adapter for conditional signals. This design significantly reduces training time and parameter consumption and supports fast sampling, such as one-step inference. The framework demonstrates strong competitive performance on both synthetic and real-world VSR datasets and achieves an excellent balance between training efficiency, inference speed, and restoration performance.

**Compliance With Llm Reviewing Policy:**

Affirmed.

**Final Justification:**

The paper is sound and clearly presented. While the originality is moderate, some design choices (e.g., the state-aware adapter inputs) and the temporal consistency analysis could be better justified; these weaknesses are relatively minor. The rebuttal adequately addressed my main concerns and improved my confidence in the method. Therefore, I have updated my evaluation and increased my score to 5.

**Key Questions For Authors:**

1. While the proposed method demonstrates strong performance on small-scale datasets, it remains unclear whether training on datasets of the same scale as those used in comparative methods would lead to substantial or only marginal improvements in performance.
2. Could the authors provide separate ablation studies on combinations of data used in the Dual-Stream Feature Projection, specifically to validate the individual contributions of different stream adaptations (e.g., $z_y$, $\hat{z}_{0, t}$, $z_t$)?
3. The proposed method yields relatively low PSNR values compared to several baselines. Could the authors offer a more detailed explanation for this observation?
4. Would it be possible for the authors to provide additional (visual) results that better demonstrate the temporal consistency and inference speed of the proposed method?

**Limitations:**

yes

**Strengths And Weaknesses:**

### Strengths

1. The authors design a framework that leverages the constant-velocity property of flow matching, achieving full-frozen DiT and time-adaptive guidance during the training process.
2. The framework uses a state-aware adapter to generate and inject conditional signals, enabling flexible training across different sampling steps.
3. The authors conduct comprehensive experiments across both synthetic and real-world datasets using multiple classes of metrics, and their method achieves strong performance across all datasets and most evaluation metrics.
4. The authors also analyze the limitations of their methods in generative restoration, such as text reconstructions, further ensuring the completeness of the method.

### Weaknesses

1. The need to incorporate both low-quality inputs and clean estimates into the state-aware adapter has not been sufficiently validated. It appears that either the clean estimate or the time-modulated tiled query alone could potentially capture the denoising state information.
2. The authors didn't provide visualization results for temporal consistency, such as tracking the same element across frames.

---

> ### Author Rebuttal · Authors · 2026-03-30
>
> **W1 & Q2: Dual-stream Ablation**
>
> We conducted three ablation studies on VideoLQ, each isolating a key component of the State-Aware Adapter:
>
> - **w/o Refinement Stream**: removes the clean estimate stream, conditioning only on the LQ input $z_y$.
> - **w/ Noisy Latent**: replaces the predicted clean estimate $\hat{z}_{0,t}$ with the noisy state $z_t$ in the refinement stream.
> - **w/o State-Aware Adapter**: replaces the entire adapter with a standard concatenation + cross-attention mechanism without time modulation.
>
> |Variant|CLIPIQA↑|DOVER↑|NIQE↓|MUSIQ↑|
> |:--|:-:|:-:|:-:|:-:|
> |w/o Refinement Stream (LQ only)|0.4603|**0.4878**|3.7782|58.64|
> |w/ Noisy Latent $z_t$|0.4570|0.4875|3.7804|58.44|
> |w/o State-Aware Adapter|0.4292|0.4663|4.3252|52.22|
> |**LiteVSR (full)**|**0.4681**|0.4846|**3.76**|**59.05**|
>
> The most significant drop comes from removing the State-Aware Adapter entirely: training becomes unstable and fails to converge within the same training iterations, resulting in large degradation across all metrics.
>
> The LQ-only and noisy-latent variants show moderate metric drops, but with notable qualitative differences. Using $z_t$ instead of $\hat{z}_{0,t}$ introduces scene-inconsistent artifacts, as the noisy latent distribution disrupts dual-stream feature interaction. Removing the refinement stream produces identical conditioning across all timesteps, leading to under-denoised regions. We will include qualitative analysis in the revision.
>
> **W2 & Q4 Temporal Consistency Analysis**
>
> We provide three pieces of evidence:
>
> **1) Supplementary Videos.** Side-by-side video comparisons with baselines were already included in the original supplementary material, demonstrating temporal consistency across multiple sequences.
>
> **2) Project Page.** We have created an anonymous project page at https://anonymous.4open.science/w/LiteVSR-website-403C/ with extended video comparisons on both synthetic and real-world benchmarks.
>
> **3) User Study.** We conducted a user study with 15 participants across 17 video sequences, covering simple, standard, and extreme degradation scenarios:
>
> |Metric|DOVE|FlashVSR|LiteVSR (Ours)|
> |:--|:-:|:-:|:-:|
> |Visual Quality|5.8%|18.7%|**75.5%**|
> |Temporal Consistency|9.3%|20.6%|**70.0%**|
> |Overall Preference|6.2%|18.3%|**75.5%**|
>
> LiteVSR is consistently preferred across all scenarios and metrics. Screenshots of the user study interface and per-scenario breakdowns are available in the User Study section of our project page.
>
> **Q1: Training data requirements**
>
> Our core contribution is a lightweight adapter that enables VSR with minimal training resources. The training datasets used by FlashVSR and SeedVR are not publicly available. To understand how our adapter behaves with different data, we selected 2000 videos from OpenVid-1M, and also trained on 50% and 75% subsets of REDS. Results on VideoLQ:
>
> |Setting|Training Data|CLIPIQA↑|DOVER↑|NIQE↓|MUSIQ↑|
> |:--|:--|:-:|:-:|:-:|:-:|
> |50% REDS|~133 clips|0.4657|0.4779|3.7581|58.53|
> |75% REDS|~200 clips|0.4709|0.4837|3.7104|59.08|
> |Full REDS|266 clips|0.4681|0.4846|3.76|59.05|
> |OpenVid-1M|2000 videos|0.3145|0.4841|5.0155|43.59|
>
> Despite 7.5× more data, training on OpenVid-1M leads to significant degradation on most metrics, suggesting that data domain relevance matters more than volume for VSR training. This highlights the advantage of our lightweight adapter design: fewer trainable parameters require less data to converge. Furthermore, we find reducing REDS to ~133 clips already achieves comparable performance to the full set, making domain-specific adaptation more practical.
>
> **Q3: PSNR Analysis**
>
> On synthetic benchmarks, LiteVSR leads on all perceptual metrics while primarily trailing DOVE and FlashVSR on PSNR. We provide visual analysis ("Understanding PSNR vs. Perceptual Quality") on our project page with side-by-side comparisons. Higher-PSNR methods produce blurred textures and unrealistic patterns (e.g., buildings, foliage), while LiteVSR generates sharper and more faithful details at the cost of pixel-level alignment, a well-known perception-distortion tradeoff [1]. Our user study (75.5% overall preference) further confirms that perceptual quality better reflects human judgment.
>
> **Q4: Inference Speed**
>
> We benchmark all methods on the same GPU across 4,771 frames:
>
> ||STAR (15-step)|DOVE (1-step)|FlashVSR (1-step)|**Ours (5-step)**|**Ours (1-step)**|
> |:--|:-:|:-:|:-:|:-:|:-:|
> |Total Time|23h 06m|2h 46m|1h 20m|4h 58m|1h 51m|
> |Sec / Frame|~17.43|~2.09|~1.01|~3.75|~1.40|
>
> Our 5-step configuration is 4.6× faster than STAR while achieving superior perceptual quality. Notably, even with single-step sampling and no distillation, LiteVSR achieves competitive quality against DOVE and FlashVSR (Table 2, 3) at comparable speed (1.40 vs. 1.01 s/frame), while FlashVSR requires 100% parameter fine-tuning and multi-stage distillation to reach single-step capability.
>
> [1] Blau & Michaeli, The Perception-Distortion Tradeoff, CVPR 2018.

---

> > ### Author Rebuttal · Reviewer_TfLy · 2026-04-03
> >
> > Thank you for the authors' response. My concerns have been addressed, and I will increase my score to 5.

---

> > > ### Author Response · Authors · 2026-04-05
> > >
> > > We appreciate your valuable feedback and willingness to raise the score. We will incorporate all suggestions in the revision.

---

### Official Review · Reviewer_5g7N · 2026-03-13

**Soundness:** 3
**Presentation:** 3
**Significance:** 3
**Originality:** 3
**Overall Recommendation:** 4
**Confidence:** 4

**Summary:**

This paper proposes LiteVSR, a lightweight method to adapt a frozen video diffusion model into one for video super-resolution. It significantly decreases the training cost by observing that the generative backbone does not need to be fine-tuned at all, and an adapter can suffice while only tuning 11% of the parameters. The adapter uses a deual stream design that takes in the low-quality input and the current denoising estimate and fuses them with a time-modulated cross attention. Parameters are not duplicated, avoiding an issue with standard ControlNet implementations. The entire system can be trained on a single A100 with just 266 REDS clips, and achieves competitive performance on standard VSR metrics.

**Compliance With Llm Reviewing Policy:**

Affirmed.

**Key Questions For Authors:**

1. Does the method perform comparably *qualitatively* on a wide range of videos outside of standard synthetic VSR benchmarks? This is the key question that needs to be answered for this paper to be convincingly accepted. Ways to address this include both a user study, a project page with good and bad examples, etc.

**Limitations:**

yes

**Strengths And Weaknesses:**

__Strengths__

1. The observation is creative and does drastically reduce the cost to training a large video super-resolution model; using only 1 gpu and 250 clips is remarkable.

2. The method is not overly complicated and is well-motivated; using zero-intiialized layers, and the adaptive unrolling strategy make sense for smoothly training the model. The state-aware adapter also makes sense to properly incorporate information from the evolving denoised estimate.

3. The ablations are very comprehensive and clearly demonstrate the importance of the various components of the method.

__Weaknesses__

1. I believe that this method produces something reasonable on synthetic VSR datasets with minimal training. However, it’s not clear without a user study and without visual examples outside standard VSR benchmarks whether this method truly matches the performance of larger trained models such as SeedVR or SeedVR2. These methods which are trained on much larger datasets and with more compute perform well on a much wider distribution of data and images, and it’s not obvious that LiteVSR does.

2. A user study is missing, which would add significantly to the paper’s value. Synthetic VSR datasets don’t tell the full story. While the metrics are reasonable, and LiteVSR is comparable to other existing methods, qualitative perception is also extremely important. A user study would significantly add to the case that LiteVSR matches the performance of other models trained with much more compute.

3. Comparisons: the comparisons should probably include SeedVR2 and OMGVSR as well as the methods already included. Furthermore, other backbones (and parameter scales, such as 7B) such as CogVideoX should be explored if possible, since only Wan was used and this would make the experiments more complete.

4. A project page with videos is really important for these types of papers. Such a website could be anonymized but a video generation related paper should really have videos that show off the method in action. Images don’t really tell the full story.

---

> ### Author Rebuttal · Authors · 2026-03-30
>
> **W4: Project page and video demonstrations**
>
> We have created an anonymous project page at https://anonymous.4open.science/w/LiteVSR-website-403C/ with extended video comparisons on both synthetic and real-world benchmarks, including side-by-side comparisons with SeedVR2. Video comparisons were also included in the original supplementary material.
>
> **W1 & W2 & Q1: User study and qualitative evidence beyond synthetic benchmarks**
>
> We thank the reviewer for this suggestion. We conducted a user study with 15 participants across 17 video sequences, evaluating visual quality, temporal consistency, and overall preference against DOVE and FlashVSR. To evaluate performance beyond synthetic benchmarks, we designed three sections:
>
> - **Standard**: 5 clips from VideoLQ, a real-world benchmark with naturally degraded videos.
> - **Simple / Extreme**: 12 videos captured by mobile phones under real-world conditions (not synthetically degraded), split by degradation severity.
>
> Videos were presented in randomized order with randomized A/B/C assignment to eliminate bias. Results:
>
> |Scenario|Metric|DOVE|FlashVSR|LiteVSR (Ours)|
> |:--|:--|:-:|:-:|:-:|
> |Overall|Visual Quality|5.8%|18.7%|**75.5%**|
> ||Temporal Consistency|9.3%|20.6%|**70.0%**|
> ||Overall Preference|6.2%|18.3%|**75.5%**|
> |Standard|Overall Preference|5.3%|36.0%|**58.7%**|
> |Simple|Overall Preference|6.6%|19.8%|**73.6%**|
> |Extreme|Overall Preference|6.6%|2.2%|**91.2%**|
>
> LiteVSR is consistently preferred across all scenarios. The advantage is especially pronounced under extreme real-world degradation (91.2% vs. 2.2% FlashVSR), demonstrating strong generalization beyond synthetic benchmarks. Screenshots of the user study interface and aggregated results are available in the User Study section of our project page.
>
> **W3: Additional comparisons and other backbones**
>
> We thank the reviewer for these suggestions. We have included visual comparisons with SeedVR2-3B in the Visual Comparisons section of our project page. We also provide quantitative comparison on VideoLQ:
>
> | Method | CLIPIQA↑ | DOVER↑ | NIQE↓ | MUSIQ↑ |
> |:--|:-:|:-:|:-:|:-:|
> | SeedVR2 | 0.2855 | 0.4323 | 4.6092 | 56.16 |
> | **LiteVSR (Ours)** | **0.4681** | **0.4846** | **3.76** | **59.05** |
>
> Regarding OMGVSR, we were unable to find a method by this name. The closest match we identified is OMGSR [1], which to our knowledge is an image super-resolution method and not directly applicable to video SR.
>
> For other backbones, our adapter interfaces with the backbone solely through zero-initialized linear injections at each DiT block, with the only model-specific component being the patch embedding initialization. HunyuanVideo and CogVideoX both satisfy this requirement and already have community ControlNet implementations (e.g., DiffSynth-Studio). We are actively implementing this and will update results if available during the discussion period.
>
> [1] Wu et al., OMGSR: You Only Need One Mid-timestep Guidance for Real-World Image Super-Resolution

---

### Decision · Program_Chairs · 2026-04-30

**Decision:**

Accept (regular)

**Comment:**

The paper received mixed scores of 3/4/5.

The reviewers agreed that the paper is technically sound, clearly presented, and proposes a compelling lightweight adaptation strategy for frozen diffusion transformers in video super-resolution, supported by comprehensive ablations. After the rebuttal, Reviewer agrh still expressed concerns about presentation issues and the fairness of the comparisons. The presentation issues can be further polished in revision, while the comparison concern is somewhat mitigated by the fact that the paper already includes six recent baselines. Overall, this weakness does not substantially undermine the paper’s main contribution.

The reconmendation is: Accept.